# Novel Methodology for Experimental Characterization of Micro-Sandwich Materials

**DOI:** 10.3390/ma14164396

**Published:** 2021-08-05

**Authors:** Samuel Hammarberg, Jörgen Kajberg, Simon Larsson, Ramin Moshfegh, Pär Jonsén

**Affiliations:** 1Department of Engineering Sciences and Mathematics, Division of Solid Mechanics, Luleå University of Technology, 971 87 Luleå, Sweden; Jorgen.Kajberg@ltu.se (J.K.); Simon.Larsson@ltu.se (S.L.); Par.Jonsen@ltu.se (P.J.); 2Lamera AB, A Odhners Gata 17, 421 30 Västra Frölunda, Sweden; Ramin.Moshfegh@lamera.se

**Keywords:** composite, Hybrix, micro-sandwich, lightweight, characterization, digital image correlation (DIC)

## Abstract

Lightweight components are in demand from the automotive industry, due to legislation regulating greenhouse gas emissions, e.g., CO_2_. Traditionally, lightweighting has been done by replacing mild steels with ultra-high strength steel. The development of micro-sandwich materials has received increasing attention due to their formability and potential for replacing steel sheets in automotive bodies. A fundamental requirement for micro-sandwich materials to gain significant market share within the automotive industry is the possibility to simulate manufacturing of components, e.g., cold forming. Thus, reliable methods for characterizing the mechanical properties of the micro-sandwich materials, and in particular their cores, are necessary. In the present work, a novel method for obtaining the out-of-plane properties of micro-sandwich cores is presented. In particular, the out-of-plane properties, i.e., transverse tension/compression and out-of-plane shear are characterized. Test tools are designed and developed for subjecting micro-sandwich specimens to the desired loading conditions and digital image correlation is used to qualitatively analyze displacement fields and fracture of the core. A variation of the response from the material tests is observed, analyzed using statistical methods, i.e., the Weibull distribution. It is found that the suggested method produces reliable and repeatable results, providing a better understanding of micro-sandwich materials. The results produced in the present work may be used as input data for constitutive models, but also for validation of numerical models.

## 1. Introduction

In the automotive industry, lighter and recyclable materials are in demand due to regulations regarding emissions and sustainability [1,2,3]. Thus, an incentive exists for innovation of lightweight materials based on, e.g., metals and composites. In addition to regulations, cost is a decisive factor when selecting materials, where cost of raw material and testing of materials and components must be accounted for. By replacing physical testing with simulations, cost can be reduced. Thus, knowledge of the material obtained by experimental characterization is vital.

Multilayered materials, e.g., laminates and sandwiches, allow for components of unique mechanical properties where each layer has been added for a specific purpose, a concept known for centuries. The result is more efficient material usage and improved strength-to-weight properties. A classic example is the use of sandwiches with balsa wood cores of the “De Havilland 98 Mosquito”, an aircraft used during the second world war [4].

The usability of sandwich materials as structural membranes and energy absorption components has been studied by several authors. Hara and Özgen [5] investigated the weight-saving potential of sandwich materials used for stiffness and damping of automotive structures. It was shown that weight savings were possible by replacing current components with appropriate sandwich materials. In the numerical studies by Ahmed and Wei [6] and Sellitto et al. [3], sandwich composites were introduced into the hood structure of a vehicle, thereby increasing pedestrian safety and reducing weight. Riccio et al. [7] studied the impact behavior of honeycomb cores based on natural fibers, using numerical models accounting for damage. The numerical models were in agreement with experiments, regarding force–displacement response and final deformed geometry. In the work by Balakrishnan et al. [8], an unsymmetrical hybrid laminate, based on glass fibers and boron steel (22MnB5), was developed and studied using three-point bending and drop tower impact tests. It was found that the heat treatment of the steel significantly affected the performance of the laminate. Further work regarding boron steel in sandwich structures has been conducted in the work by Hammarberg et al. [9]. A sandwich with a bidirectionally corrugated core was manufactured and evaluated in three-point bending. Energy absorption was studied by Schneider et al. [10], where corrugated sandwich beams, made from self-reinforced polyethylene terephthalate, were manufactured and tested under dynamic loading. The sandwich material of the study was found to be competitive when compared to sandwich panels based on, e.g., aerospace aluminum. Numerical models were created, able to reproduce the response of the experiments. In the work by Hammarberg et al. [11], a numerical investigation was conducted, studying the energy absorption potential of ultra-high strength steel (22MnB5) sandwich materials with perforated cores. Failure was accounted for by including a stress state-dependent damage model. The sandwich concept indicated an increase in specific energy absorption, as compared to a reference component of equivalent weight, based on solid steel. Najibi et al. [12] conducted a numerical study of tubes filled with functionally graded foam, subjected to axial compression. Various density distributions of the foam were investigated in order to optimize the specific energy absorption. Utilizing sandwich materials offers great potential due to the possibility of combining various core and skin materials, allowing tailored properties, and meeting demands regarding structural stiffness and energy absorption.

More recently, the sandwich concept has been adopted using much thinner layers, generating so-called micro-sandwich (or thin sandwich) materials, suitable for forming operations. An early version of a micro-sandwich was developed by Bhart et al. [13], where the cell walls of a honeycomb pattern consisted of a micro-sandwich structure. Micro-sandwich materials typically consist of metal skins with a polymeric core, see, e.g., in [14,15,16]. Several micro-sandwiches have been presented in the literature, e.g., Hylite [17]. As reported by Kim et al. [18], Burchitz et al. [19], Carrado et al. [20], Hylite exhibits excellent stiffness properties and is formable using traditional forming methods, e.g., drawing [21]. Other such micro-sandwiches are Thyssenkrup’s bondal (see the work by Kami et al. [22]) and Litecor (see the work by Tanco et al. [15]), with focus on vibration damping and lightweight design, respectively. Micro-sandwiches with fibrous cores, where the fibers are oriented along the thickness direction of the sandwich, are also available [23,24]. Hybrix^TM^ is such a micro-sandwich, developed by Lamera AB. The fibrous core consists of microscopic fibers, bonded to the skins by an epoxy resin, see Figure 1a.

Mechanical characterization of micro-sandwiches has been carried out by several authors. In the work by Kim et al. [18], a sandwich consisting of aluminum skins and a polypropylene core was studied with respect to strain hardening, strain rate sensitivity, and anisotropy. Mechanical properties of the sandwich were obtained using uniaxial tensile tests. The formability of the sandwich was evaluated using punch-stretching tests from which a forming limit diagram (FLD) was obtained. Kim et al. [18] concluded that better understanding of the plastic region of sandwich sheets is necessary. A similar sandwich was studied by Parsa et al. [25] where the FLD was obtained numerically, using finite element simulations together with a damage model. The numerical model was validated against experimental data and good agreement was found. Sokolova et al. [26] studied a micro-sandwich with skins of stainless steel and a polymer core. Tensile and deep drawing tests were conducted and the sandwich was found to have great forming potential. Liu and Xue [27] studied micro-sandwiches based on polyethylene cores and aluminum skins. Core thickness was varied to investigate ductility of the micro-sandwich, using the limiting dome height test. It was found that the sandwich had improved properties as compared to monolithic aluminum with respect to elongation at fracture, limiting dome height and forming limits. Logesh and Bupesh Raja [28] investigated how the skin material affected formability of a micro-sandwich with a polypropylene core. Formability was evaluated using Erichsen cupping test, and the authors conclude that the sandwich material could be used for automotive parts, such as body panels and dashboards. Characterization of Hybrix^TM^ was conducted in the work by Pimentel et al. [29]. The mechanical properties of the Hybrix^TM^ core were determined indirectly by performing separate tensile tests for the Hybrix^TM^ sheet and the skins. By deducting the response from the skins, the in-plane mechanical properties of the core were obtained. Such a methodology requires a set of assumptions, e.g., that the volume remains constant during deformation [22]. From Figure 1a, it should be clear that the core has limited contribution to the in-plane properties of the Hybrix^TM^ sheet, as the fibers are in the direction of the thickness with the purpose to maintain the initial distance of the skins.

Research regarding modeling of sandwich material has been conducted by several authors, suggesting layered finite element shells for representing the sandwiches, see, e.g., in [30,31,32,33]. These have been benchmarked against commercially available shell formulations with good agreement. Modeling of the micro-sandwiches has mainly been done based on in-plane tensile tests, ignoring transverse properties of the core, see in [29,34]. As stated by Pimentel et al. [34], there is a lack of knowledge regarding the mechanical behavior of the micro-sandwich, limiting its use in the automotive industry. To further advance the use of micro-sandwich materials, experimental methods for characterization is on demand. Robust methods for characterization of mechanical properties offer a solid foundation from which numerical models receive their input data, allowing further development and validation of constitutive models. Thus, it is necessary to further develop methods for mechanical characterization, studying the mechanical properties of the micro-sandwich and how the constituents interact during loading.

Due to the heterogeneous and anisotropic characteristics of composite materials, their mechanical properties generally exhibit intrinsic statistical dependencies [35]. Thus, statistical methods are necessary for understanding and characterizing such materials. The Weibull distribution [36,37] has been applied by several authors for this purpose [35,38,39], describing statistical behavior of materials, including pellets [40], particles [41] ceramic composites [42], and glass materials [43,44]. Due to its simple mathematical form and its adaptability to experimental data, the Weibull distribution has been preferred over other distributions and used by several authors Madjoubi et al. [43].

In the present work, a novel experimental methodology is developed for characterizing mechanical properties of micro-sandwich materials and in particular their cores. The work is motivated by the need for reliable, robust methods for determining mechanical properties of micro-sandwich materials, enabling numerical models able to predict damage, failure and delamination during manufacturing of components, e.g., cold forming. The methodology is applied to the micro-sandwich Hybrix^TM^, developed by Lamera AB. Hybrix^TM^ is an umbrella trademark for advanced lightweight materials where a variety of combinations of cores and skins are possible. The novelty of the presented work lies in the direct characterization of the out-of-plane mechanical properties of the core, where a statistical method is applied to describe its variable nature. Furthermore, digital image correlation (DIC) is utilized to control and ensure that pure tension or shear stress states are present during testing. Thus, five load cases (LCs) are used in the experimental study, including tension (and compression) in the thickness direction, i.e., normal to the sandwich, and transverse shear, from which the properties of the core are derived. A dispersion was noted for the obtained data. Thus, the Weibull distribution function is used to characterize the variation of material properties. From the Weibull distribution, the standard deviation and mean are obtained for the material properties analyzed in the study.

## 2. Materials and Methods

The experimental work of this study was aimed at developing an experimental methodology for characterizing mechanical properties of micro-sandwich materials and in particular their cores. In the present section, the materials of the study are presented as well as the experimental procedure.

### 2.1. Materials

In the present study, the micro-sandwich material Hybrix^TM^, CC030030PC-120 TMK-T7609, was investigated. A conceptual image of Hybrix^TM^, together with microscopic photographs of one of the test specimens, are presented in Figure 1. Three layers made up the sandwich: two carbon steel skins bonded by an adhesive to a fibrous, polyamid core with fibers preferably aligned in the thickness direction of the sandwich, applied using electrostatic flocking. The thicknesses of skins and core were 0.30 mm and 0.60 mm, respectively. Material data for the constituents of the Hybrix^TM^ sandwich, i.e., the bulk material data, are summarized in Table 1.

### 2.2. Experimental Procedure

In this section, the experimental procedure suggested for characterizing micro-sandwich materials is presented. Note that the fibrous core and the adhesive, binding core and skins, was treated as an equivalent material. Thus, at this stage, no distinction is made between core and adhesive. To ensure the integrity of the micro-sandwich materials during service life, delamination must not occur nor should the core collapse. Furthermore, it was assumed that due to the orientation of the fibers in the core, their contribution to in-plane stiffness was negligible. Thus, the aim of the experimental procedure was to determine the out-of-plane mechanical properties of the core, i.e., tensile and compressive properties normal to the sandwich as well as shear rigidity. Furthermore, the core was also subjected to mixed-mode stress states influenced by both normal and shear stress.

Five LCs were used to study the mechanical response of the specimens: LC 1: tensile normal to the sandwich, LC 2: compression normal to the sandwich, LC 3: out-of-plane shearing, LC 4: off-axis loading 30°, and LC 5: off-axis loading 60°. LC 1–3 correspond to data which can typically be used as input for various numerical, constitutive models. The response of LC 4–5 may be used for validation, as they produce stress-states combining tension and shear, or as input data for models where off-axis loading may be prescribed. A conceptual illustration of the five LCs is presented in Figure 2 and in Table 2 the LCs are summarized.

To impose LCs 1 and 3–5, four test tools were manufactured, see Figure 3. A detailed presentation of the test tool used for LC 1 is given in Figure 4 with all constituents presented. Each test tool was milled from a single steel piece. Guides were mounted in the holes on the lower part of the test tools by shrink-fitting and Teflon bearings were fitted in the upper holes, minimizing friction during testing. For LC 3, i.e., shearing, the guides were removable, and thus removed before testing. Test specimens of size 15 × 40 mm^2^ were obtained by water jet cutting, see Figure 5. The micro-sandwich specimens were prepared and bonded to the test tools according to Section 2.3. A torque of 5 Nm was applied to the nuts of the guides, creating a small pressure during curing of the bonds. Spacers were placed around the guides, between the test tools to ensure the integrity of the specimen was not compromised by the pressure applied. A servo-hydraulic testing machine, Instron 1272, with a 30 kN load cell was used for LCs 1 and 3–5, see Figure 6a. For LC 2, i.e., compression, a 100 kN load cell was used, due to the forces required to compress the specimens. The experimental setup presented in Figure 6b. For all LCs force and displacement were measured by the internal loading cell of the testing machines and a mechanical extensometer, respectively. All LCs used a displacement-controlled loading rate of 0.2 mm/min. Additionally, digital image correlation (DIC) [45] was used, partly to qualitatively evaluate displacement of the epoxy, bonding specimens to the test tools, for LCs 1 and 3–5 but also to study the fracture mode of the sandwich core. The DIC measurements were carried out using a GOM Aramis system (version 6.1). A CCD camera was used with a spatial resolution of 2048 × 2048 pixels, using a macro lens with a focal length of 50 mm and maximum aperture of 1:2.8. During the compressive testing, the micro-sandwich specimens were placed between two steel cuboids, allowing better illumination of the micro-sandwich specimen. A summary of the parameters to be determined in the experimental study is presented in Table 3.

### 2.3. Specimen Preparation

After water jet cutting, edges caused by the cutting were removed by grinding. Before joining the specimens to the test tools, the specimens were cleaned with acetone, sandblasted, and cleaned again. Araldite 2015-1, a two-component epoxy, with a tensile strength of 30 MPa, a tensile modulus of 2 GPa, and a fracture strain of 4.4%, was used to join the specimens to the test test tools. After specimens had been mounted on the test tools, epoxy residue on the edges of the specimen was removed. Test tools and specimens were placed in a furnace, curing the epoxy at 40 °C for 16 h. After curing, the area of test tools and specimens to be analyzed using DIC, received a final grinding and sandblasting, removing any remaining epoxy residue. Then the speckle pattern was applied to the specimen surface and the test tool was mounted in the testing machine. The work flow is summarized in Figure 7.

## 3. Statistical Analysis: Weibull Distribution

To analyze the variation of the mechanical properties of the micro-sandwich, statistical analysis was indispensable. A two-parameter Weibull distribution [36] was fitted for each parameter in Table 3. The probability density function (PDF), f(x), and cumulative probability of failures, F(x), typically referred to as the cumulative distribution function (CDF), are given as
(1)$$f=\frac{m}{x}{\left(\frac{x}{x_{0}}\right)}^{m}e^{-{\left(\frac{x}{x_{0}}\right)}^{m}}$$
and
(2)$$F\left(x\right)=1-e^{-{\left(\frac{x}{x_{0}}\right)}^{m}}.$$

The PDF, Equation (Equation 1), contains the probability of a test specimen equaling a certain value of x≥0, whereas the CDF, Equation (Equation 2), contains the probability of a test specimen equaling a certain value of *x* or lower. The parameters $x_{0}$ and m are parameters typically referred to as the scaling parameter and Weibull modulus, respectively. A statistical mean, $M_{WB}$, for a parameter *x* of the Weibull distribution, is given by
(3)$$M_{WB}\left(x\right)=x_{0}\Gamma \left(1+\frac{1}{m}\right)$$
where Γ is the gamma function. The variance for the Weibull distribution is determined by
(4)$$V_{WB}=x_{0}^{2}\left[\Gamma \left(1+\frac{2}{m}\right)-\Gamma^{2}\left(1+\frac{1}{m}\right)\right]$$
which equals the square of the standard deviation. To fit the Weibull distribution against experimental data, Equation (Equation 2) was subjected to a double logarithmic transformation, resulting in
(5)$$\ln\left(\ln\left(\frac{1}{1-F\left(x\right)}\right)\right)=m\cdot \ln\left(x\right)-m\cdot \ln\left(x_{0}\right).$$

By plotting $\ln\left(\ln\left(\frac{1}{1-F\left(x\right)}\right)\right)$ and $\ln\left(x\right)$ as the dependent and independent variables, respectively, for the experimental data, an approximately straight line was fitted. From the equation of the fitted line, the Weibull modulus, *m*, corresponds to the slope of the line, and the scaling parameters, $x_{0}$, may be solved for. *F*, in Equation (Equation 5), is typically determined using the median rank method. Several expressions have been suggested for approximating the failure probability, F(x), [35]:(6)$$F_{i}=\frac{i}{n+1}$$
(7)$$F_{i}=\frac{i-0.5}{n}$$
(8)$$F_{i}=\frac{i-0.3}{n+0.4}$$
where *n* corresponds to the total number of specimens. By sorting the experimentally determined values of *x* in ascending order, *i* corresponds to the ordered specimen number, i.e., *i* is a vector of integers i=1,2,3…n−2,n−1,n, with i=1 and i=n corresponding to the lowest and highest values of *x*, respectively. It has been shown that Equations (Equation 7) and (Equation 8) produce results with higher accuracy, as compared to using Equation (Equation 6) [46]. For the present work, Equation (Equation 7) was used. By ranking the specimens in ascending order (from lowest to highest) and computing $F_{i}$ for each specimen, Equation (Equation 5) was used, producing an approximately straight line for the specimens, from which the Weibull parameters were estimated. From the Weibull distribution, standard deviation and mean were computed. The arithmetic mean corresponds to the expected value of the mechanical properties, and approaches the statistical mean of the fitted distributions as the sample size increases. Thus, the arithmetic mean was compared against mentioned distributions.

## 4. Results and Discussion

This section consists of two major parts: In the first part, the mechanical properties obtained during the experimental study are presented and discussed. In particular, the out-of-plane properties of the micro-sandwich were studied. At least 15 specimens were studied for each parameter. LCs 1–3 correspond to states of pure stress and strain, whereas LCs 4 and 5 contain mixed-mode stress states involving normal as well as shear stress. Thus, for LC 1–3, the response is presented as engineering stress and strain, whereas force and displacement are used for LC 4 and 5. The second part deals with the statistical analysis of the obtained data where statistical distributions are fitted to the experimentally obtained data.

### 4.1. Mechanical Properties of Hybrix^TM^

In order to ensure reliable and repeatable results, several specimens were subjected to each LC, necessary for the statistical analysis. The specimens were bonded to the test tools, corresponding to the LC, by an epoxy. A requirement is thus that the epoxy does not fail during testing (cohesive or adhesive failures). The lap shear strength and tensile strength of the epoxy were listed by the manufacturer at approximately 30 MPa and the fracture strain was given as 4.4%. By studying both the specimens after loading as well as the DIC footage, it was found that the performance of the epoxy was not a limiting factor except for in LC 1, where four distinct failure modes were observed (or a combination of the four), presented schematically in Figure 8, where the skins have been omitted to simply the illustration. Figure 8a,b corresponds to the performance of the bonding, where cohesive and adhesive failures are presented, respectively. The cohesive failure typically resulted from a lack of epoxy in the joint, whereas the adhesive failure may be a result of poorly prepared and/or contaminated surfaces. From the DIC images of LC 1, two types of core failures were observed: core tearing and brittle core failure, see Figure 8c,d. A difference of the force–displacement response between the tearing and the brittle failure was noted, where the former caused a ductile force–displacement response and the latter a more abrupt decrease in loading after the peak force, i.e., a more brittle failure behavior. The DIC footage of the test specimens was used to sort between the tests where the epoxy did not fail. Specimens exhibiting failures in accordance with Figure 8a–c were excluded from further analysis and were thus not used for determining the mechanical properties of the Hybrix^TM^ core. Combinations of the failure modes were also observed, see Figure 9, where both a core fracture and an adhesive failure has occurred. Due to the various combinations of failure modes, approximately 30% of the specimens for LC 1 produced usable results.

A comparison between the measured DIC strain fields for tearing and brittle failure is presented in Figure 10. In Figure 10b, strain is clearly localized near the left edge of the core, resulting in a tearing fracture. In Figure 10d, an evenly distributed strain field is observed for the core, causing a brittle failure. In Figure 10e, the measured area used for DIC is presented.

Note that for LCs 2–5 no distinction could be made between various failure modes of the core, and specimens were only disregarded if a cohesive or adhesive failure occurred, which was rare.

#### 4.1.1. Load Case 1: Tensile Loading Normal to the Sandwich

In Figure 11a, the tensile engineering stress–strain responses for 15 selected specimens of the Hybrix^TM^ micro-sandwich are presented. The stress–strain curves are presented up until the maximum stress, marked with black markers, after which a rapid, brittle fracture occurred, marked with red markers. A certain variation of the response was expected due to the varying distribution of fiber density and orientation. The highest maximum stress was found to be larger than the lowest maximum stress by a factor of two. The maximum stresses are presented in Figure 11b. The equivalent Young’s modulus, presented in Figure 11c, was computed as the mean slope of the initial part of the stress-strain response, for stresses fulfilling $0.15\sigma_{max}\leq \sigma \leq 0.3\sigma_{max}$. Compared to the maximum stresses, the equivalent Young’s modulus was found to be more closely grouped together. Still, for both parameters the variability is evident, which may be expected due to the stochastic nature of the core.

DIC was used to characterize the fracture modes of both the epoxy and the core. The strain field for one of the specimens in the study is presented in Figure 11. The speckle pattern is presented in Figure 11d, and in Figure 11e,f, the strain field along the loading direction is presented before and after fracture, respectively. Note two things here: the strain field is approximately evenly distributed over the width of the specimen and the strain is concentrated to the core, see Figure 11e. Thus, the strain of the epoxy, test tools, and skins is negligible. This is an important observation, as the displacement was measured by an extensometer attached to the test tool, not the specimen. Regarding the epoxy, note that even if the epoxy would reach its failure strain of 4.4% (as stated by the manufacturer), its contribution to the displacement would be negligible due to its thickness which was assumed to be orders of magnitude smaller than the core’s thickness. In Figure 11f, the fracture is presented which occurred simultaneously over the specimen width. This agrees with the stress–strain response in Figure 11a, where an abrupt drop in stress was seen. In Figure 12, two specimens that exhibited a brittle core failure are presented. Regarding the core fracture, the failure may either be a pure adhesive failure, a pure fiber fracture, or a combination. In Figure 12, it is seen that a combination of failure modes has occurred since the specimens have fibers remaining on both skins and patches of missing fibers are also noted. Similar fractures were observed for all specimens subjected to LC 1.

#### 4.1.2. Load Case 2: Compressive Loading Normal to the Sandwich

All specimens were loaded in compression to a maximum force of approximately 95 kN at which point the test was terminated, due to the limits of the loading cell. A typical loading response for the test specimens in uniaxial compression is presented in Figure 13. Initially, a nonlinear loading response was observed, assumed to be due to the specimen and machine not being in full contact. When full contact was reached, at ϵ≈0.03, the stress was proportional to the strain and a linear-elastic material response was assumed. The upper limit of the linear elastic regime was determined by computing the second derivative, $f^{\prime \prime }\left(\epsilon \right)=\frac{d^{2}\sigma }{d\epsilon^{2}}$, of the stress–strain response. The first point at which $f^{\prime \prime }\left(\epsilon \right)<0$ was used as the end of the linear elastic response and the slope of the stress–strain at this point was used as the equivalent Young’s modulus in compression. The initial nonlinear response of the specimens was replaced by extrapolating from the final point of the elastic region back down to zero using the equivalent Young’s modulus, see Figure 13b. A similar procedure was used for all specimens and the resulting stress-strain response is presented in Figure 14a.

The response shown in Figure 14a exhibits similar characteristics to what is observed for cellular/porous materials with a high relative density, see [47,48,49]. This is reasonable since the core consists of fibers and cavities. Typically, the response of porous materials is divided into three regions. At initial load, the response follows a straight line, corresponding to the equivalent Young’s modulus of the cellular material. This behavior can be seen in Figure 14a for 0≤ϵ≤0.05. As the load increases, the cellular material starts crumbling, due to plastic buckling of the cell walls, causing a plateau in the response curve. How distinct the plateau is depends on the relative density of the material. A more distinct plateau is generally observed for materials with lower relative densities. As can be seen in Figure 14a, a plateau, similar to what is observed for high-density foams, is present for strain values between 0.05 and 0.15, as fibers are compacted and force is not transferred in an effective manner between the two skins. Due to the high relative density, the plateau is not as prominent as for materials of lower relative densities. As the fibers starts to crumble and the relative density increases, the core starts transferring the load again between the skins, and the load starts rising again. This can be seen in Figure 14a for ϵ≥0.15. From this point and onward, the load keeps on rising, the relative density approaches a value of 1, and the slope of the stress–strain curve approaches a slope equal to the stiffness modulus of the bulk material.

Due to the scatter of the response, the stress–strain curves are presented up until a strain of 0.25 and the corresponding stresses are presented in Figure 14b. The equivalent Young’s modulus is presented in Figure 14c. Note that in contrast to the other LCs, no epoxy is present during compression. Thus, the spreading is directly derived from a variety in the core material.

The strain field for one of the specimens, obtained using DIC, is presented in Figure 14e. Regions of red and blue correspond to approximately zero strain and maximum compressive strain, respectively. The major part of the compressive strain occurs in regions where the specimen is in contact with the cuboids, motivating the removal of initial part of the compressive stress–strain response, as well as in the specimen core. Note that large areas of the test specimen exhibits strains approximately equal to zero strain (blue regions). This may be due to boundary effects on the measured surface, caused by the edge of the cuboids and specimen not being in perfect contact as well as cavities due to the fiber distribution.

#### 4.1.3. Load Case 3: Out-of-Plane Shear Loading

In Figure 15a, the stress–strain responses of 20 specimens, subjected to shearing, are presented. The variety of the response was evident, with the highest peak stress being larger than the lowest peak force by a factor close to two, see Figure 15b. Peak stresses and maximum strains are marked with black and red markers, respectively, in the Figure 15a. The response presented in Figure 15a is clearly more ductile than LC 1. Commonly, after the specimen has reached its peak stress, damage in the core region grows, reducing the load resistance capacity until a total failure occurs. Due to the ductile response, the core is able to bear load well beyond the peak stress, a major difference between LC 1 and 3. The equivalent shear modulus, determined in a similar manner as the equivalent Young’s modulus for LC 1, exhibited a wide spread, see Figure 15c.

DIC was used to evaluate displacements and strains during loading, to ensure no significant movement occurred in the epoxy. The speckle pattern is presented in Figure 15d, whereas Figure 15e,f presents the strain field at max load and fracture, respectively. The shear strain is mainly concentrated to the core of the micro-sandwich and strains occurring within the epoxy, test tools and skins are negligible. It should be noted that, even if the shear strain of the epoxy layers was of the same magnitude as for the core, it would only have a limited influence on the measured displacement by the extensometer due to the thickness of the epoxy, being orders of magnitude smaller than the micro-sandwich core. In Figure 15f, it is seen that the fracture is concentrated to the core of the micro-sandwich. In accordance with the ductile load response for shearing, it is also noted that fracture is not abrupt and the cracks grow over time as the displacement is increased. The fractured cores of two specimens are presented in Figure 16. As for the fractured cores presented for LC 1, a mixed-mode fracture is observed, where both epoxy and core fibers seem to have failed. This type of fracture was representative for specimens subjected to shearing.

#### 4.1.4. Load Case 4: 30 Degrees Off-Axis Normal

In LC 4, specimens were subjected to a 30° off-axis loading, measured from the normal of the sandwich. The obtained force–displacement curve is presented in Figure 17a with the maximum force and displacement indicated with black and red markers, respectively. Maximum force is also presented in Figure 17b. A large variation is seen from the force–displacement response and both brittle and ductile fractures were observed. In Figure 17a, brittle fractures are identified as curves where maximum and fracture force occurs at approximately equal displacements, similar to LC 1. A condition which seems to be fulfilled by at least five curves. Remaining curves, exhibits a ductile response, similar to LC 3. Stiffness, obtained from the slope of the initial part of the force–displacement curves, is presented in Figure 17c. A similar variation as LCs 1–3 is observed, where the highest stiffness is larger than the lowest stiffness by a factor of approximately three.

The strain field during loading was obtained using DIC. In Figure 17d the speckle pattern is presented and Figure 17e,f correspond to the strain field before and during fracture, respectively. The strain field is presented for one of the specimens exhibiting a ductile response. From the strain field it is clear that the majority of the strain occurs within the core of the micro-sandwich. Thus, displacements occurring within the epoxy, test tools and skins are negligible for LC 4. Figure 18, contains two specimens subjected to LC 4. In accordance with LC 1 and 3, the core failure is mix of fiber breakage and adhesive failure.

#### 4.1.5. Load Case 5: Loading 60 Degrees

In Figure 19a, the force displacement for 20 specimens subjected to an off-axis loading of 60° is presented. Maximum and failure forces are indicated in the figure. A variation is observed with regard to the maximum force, see Figure 19b. In a similar manner to LC 4, the test specimens exhibited semi-brittle and ductile fracture. For the present LC, a semi-brittle response is defined as force–displacement curves where displacements corresponding to maximum force and failure force are separated by a factor of approximately two. In general, the force–displacement response exhibited a more ductile response as compared to LC 4. This is not surprising as LC 5 should be closer to shearing than LC 3. The stiffness, computed as the slope of the initial part of the curves in Figure 19a, and presented in Figure 19c.

DIC was used to analyze strain during loading. The speckle pattern is presented in the Figure 19d. In Figure 19e,f, the strain fields before and during fracture are presented. The strain fields indicate that the majority of strain occurs within the core for LC 5. Thus, displacements in the epoxy, test tools and skins are negligible. The fracture of the specimens subjected to LC 5 are similar to what has been presented previously where a mix of fiber breakage and adhesive failure appears to have happened. This is presented in Figure 20.

### 4.2. Statistical Analysis

In this section the results presented in Section 4.1 are analyzed using statistical methods, in accordance with Section 3. In particular, a two-parameter Weibull distribution was fitted against the parameters presented in Table 4. For each parameter at least 15 test specimens (observations) were used.

From Equations (Equation 5) and (Equation 7), the linearized relationship between the probability, *F*, and the equivalent Young’s modulus in tension, $E_{t}$, was obtained. By fitting a straight line to the linearized data, the scaling parameter, $x_{0}$, and shape parameter (or Weibull modulus), *m*, for the Weibull distribution of the equivalent Young’s modulus were determined. In an equivalent manner, the linearized relationship and the Weibull parameters were determined for the remaining parameters of Table 4, presented in Figure 21. Observe that the slope of the fitted straight line is equivalent to the Weibull modulus and that a higher Weibull modulus corresponds to less scattering of the observed data, i.e., a steeper slope of the fitted straight lines presented in Figure 21. It is worth noting the difference in the Weibull modulus for the equivalent Young’s modulus in tension (LC1) and compression (LC2), see Figure 21a,c. A much higher Weibull modulus, i.e., slope of the fitted straight line, is observed for tension compared to compression, despite the fact that LC1 introduces additional parameters (or uncertainties), due to the epoxy used for joining specimens to the test tools and the presence of guide pins used to control the movement during loading. In LC2, the specimens were placed between the test tools and a compressive load was applied. Still, a greater variation, i.e., lower Weibull modulus, is observed in compression as compared to tension, indicating that the dispersion of the response may be due to the material and not the method of testing. A similar observation, though not as distinct, can be made for the Weibull modulus of the stresses $\sigma_{t,max}$ and $\sigma_{c,max}$ in Figure 21b,d. Regarding LC3, i.e., shearing, the equivalent shear modulus exhibits a Weibull modulus similar to compression, indicating a greater dispersion in the data, whereas the maximum shear stresses are much more closely grouped together, see Figure 21e. Regarding LC4 and 5, a great dispersion is observed for both stiffness and maximum force, see Figure 21g–j. This was expected due to variations observed in the force-displacement response.

The R^2^ value, i.e., goodness-of-fit, indicates how well the Weibull distribution fits the data. An R^2^ value of one, would indicate a perfect fit. As can be seen in Figure 21, the $R^{2}$ value is above 0.8 for all cases, indicating adequate fitting.

With the Weibull modulus, *m*, and scaling parameter, $x_{0}$, determined, the corresponding PDF for each parameter in Table 4 was obtained, see Figure 22. The mean and standard deviation are presented for each parameter, computed according to Equations (Equation 3) and (Equation 4). To further illustrate how well the Weibull distribution fits the data, experimental data points and the CDFs are plotted in Figure 23. The Weibull distributions agree well with the experimental data.

Furthermore, Weibull mean stress–strain curves were computed, based on the data presented in Figure 11a, Figure 14a and Figure 15a. For each value of strain, the corresponding stresses were used to fit a Weibull distribution from which the statistical mean and standard deviation were obtained. The result is presented in Figure 24. In the figure, the error bars indicate the standard deviation for each computed value.

### 4.3. Summary of Results and Discussion

In Table 5, stiffnesses and maximum loads are summarized for each LC. A relevant question is if there exists a correlation between stiffness and maximum load, which was investigated by computing correlation coefficients for the corresponding parameters, see Table 5. For LC1 and LC2, correlation coefficients of 0.26 and 0.37 were obtained, respectively, indicating weak correlations. Stronger correlations were found for the remaining LCs, LC 3–5, see Table 5. Thus, loading the test specimens along the thickness direction, i.e., fiber direction, seems to produce a greater variability between stiffness and maximum load, as compared to the LCs with influence of shear.

The presented results contained a scatter in the mechanical properties of the micro-sandwich, which may be derived from at least two sources: variations introduced during manufacturing and errors from the experimental method. Note that as such variations were also present during the compressive testing, a reasonable deduction is that a major part of the variability is due to the material itself, not just the testing method. However, it is of importance to discuss errors that may have been introduced during testing as well.

The first source to be considered is the manufacturing method of the micro-sandwich. As the fibers are distributed using flocking, a spatial variation in relative density and fiber orientation is inevitable, see Figure 1b,c. The variation of fiber orientation may also affect bonding strength between core and skin, as the mounting depth of the fibers in the adhesive of the core will vary. These variations will most likely have an effect on the load response and mechanical properties of the material. The magnitude of this effect is, however, not determined, and as the test specimens increase in size, it seems reasonable that the variations would be smoothed out and vanish. Conducting test series, using much larger test specimens, would thus be of interest to investigate if the variations can be minimized.

A second source is errors related to the testing method including alignment errors when mounting the test tools in the test machine and uneven layers of adhesion when the specimens were bonded to the test tools. Such alignment errors would likely cause uneven stress distributions of the micro-sandwich and thus underestimate the loading response, the effective area would be less than the assumed 15 × 40 mm^2^. However, it seems reasonable that this would cause tearing of the core, at least for LC1, and would thus be identifiable using DIC, depending on the direction of the crack propagation. A variation of the effective area between specimen would also influence the results, as, e.g., a one millimeter reduction in length and width would reduce the area by 9%. A homogeneous, average sandwich thickness was used in this work. As the thickness varied over the specimens, this could also be causing stress concentrations during testing, reducing the effective area. Furthermore, evaluation of fracture was limited to studying a single edge of the specimen. To further study these effects, such as fracture, alignment errors and effective area, it would be beneficial to use three dimensional scanning methods during testing, from which the loaded fibers could be studied in greater detail.

## 5. Conclusions

Micro-sandwich materials possess many interesting characteristics, but may be challenging to work with due to their dimensions. This work presented a novel method for mechanical characterization of micro-sandwich cores including statistical analysis, allowing for a better understanding of each constituent of micro-sandwich materials. The novelty of the work lies in the characterization of the out-of-plane mechanical properties of micro-sandwich materials, using the developed test tools of the present work together with DIC for evaluating failure modes and statistical methods for analyzing the test data. The suggested method is a stepping stone for experimental procedures to better predict out-of-plane properties of micro-sandwich cores.

From the mechanical testing of Hybrix^TM^, a variation was observed from the response. As the greatest variation was observed during compression testing, it was concluded that the variability is mainly due to the material and not the experimental method. This conclusion was further strengthened by the fact that the majority of strain, measured using DIC, occurred within the core. This implies that the test tools are mainly subjected to a rigid body displacement, and displacements measured on the test tools are approximately equal to those of the core. Thus, the suggested methodology provides a means for determining the mechanical properties of micro-sandwich cores in a repeatable manner. The methodology can thus be adapted for other materials, not studied in the present work. The methodology relies on joining tests specimens to test tools, by an epoxy. The test tools are clamped in the testing machine using pneumatic grips. Thus, a limiting factor of the test setup is the strength of the core material and the epoxy. Different epoxy strengths are available, but may require elevated curing temperatures which may degrade the performance of the core material.

When performing tensile tests normal to the sandwich thickness, i.e., provoking delamination, it was found that high demands were placed on specimen preparation, epoxy, and experimental setup as small alignment errors may cause stress concentrations, causing an adhesive failure. By performing the test preparation according to this work, ensuring a controlled, adequate thickness of the epoxy, the performance of the bond can be improved.

Due to the wide spread of the response seen from the micro-sandwich material of the study, is necessary to utilize statistical methods for determining means and measuring the spreading of the data. Furthermore, it seems inadequate to only rely on means, due to the wide spread of the data, especially if numerical models are to be used. Instead, numerical models should take into account the statistical distributions which may be obtained from the results presented in the work.

This work contributes by suggesting an experimental test method for studying the out-of-plane properties of micro-sandwich materials. The information obtained using this method can be applied to further develop micro-sandwich materials and as input data for constitutive modeling. A better understanding of micro-sandwich material is necessary for integration into the industry and this work provides a means for achieving this goal.

## Figures and Tables

**Figure 1 materials-14-04396-f001:**
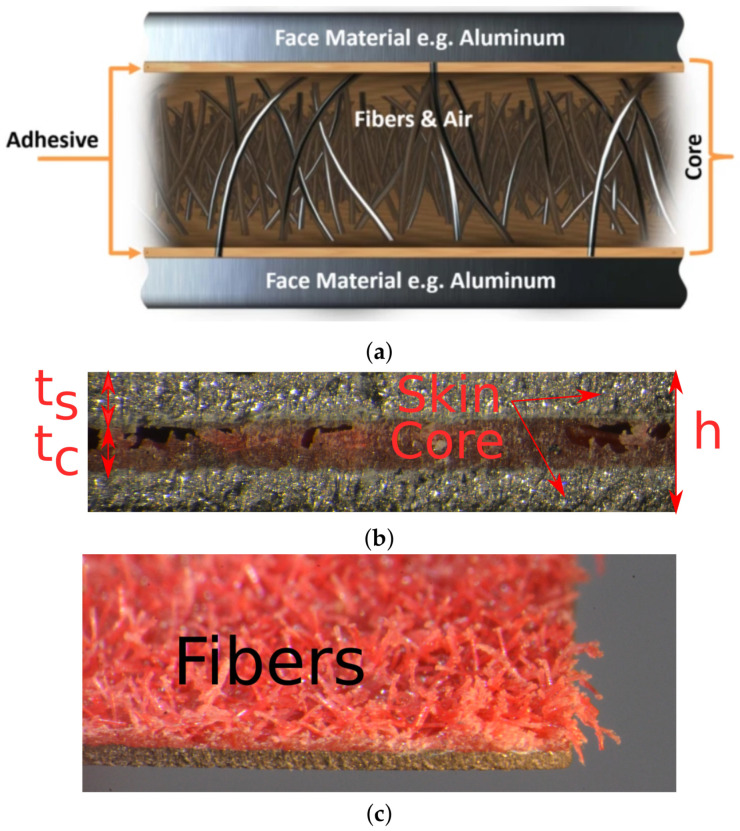
(**a**) Conceptual description of the Hybrix^TM^ sandwich and its constituents; (**b**) microscopic photograph of Hybrix^TM^ with a skin and core thickness of 0.3 mm and 0.6 mm, respectively; and (**c**) fibrous core is presented after the removal of the upper skin.

**Figure 2 materials-14-04396-f002:**
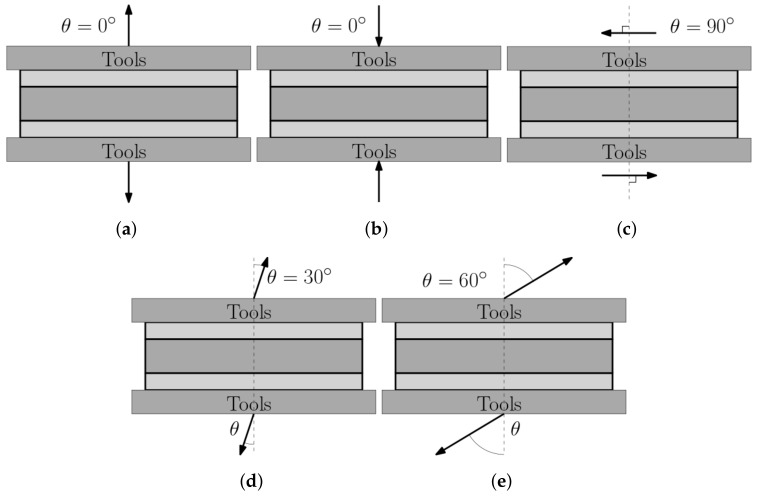
A conceptual illustration of the five LCs used for mechanical characterization of the micro-sandwich core are presented. Arrows indicate the direction of displacement. In (**a**–**c**) LCs 1–3 are depicted, corresponding to homogeneous, equivalent states of stress. In (**d**,**e**), LCs 4 and 5 are presented, corresponding to combinations of tensile and shear stresses.

**Figure 3 materials-14-04396-f003:**
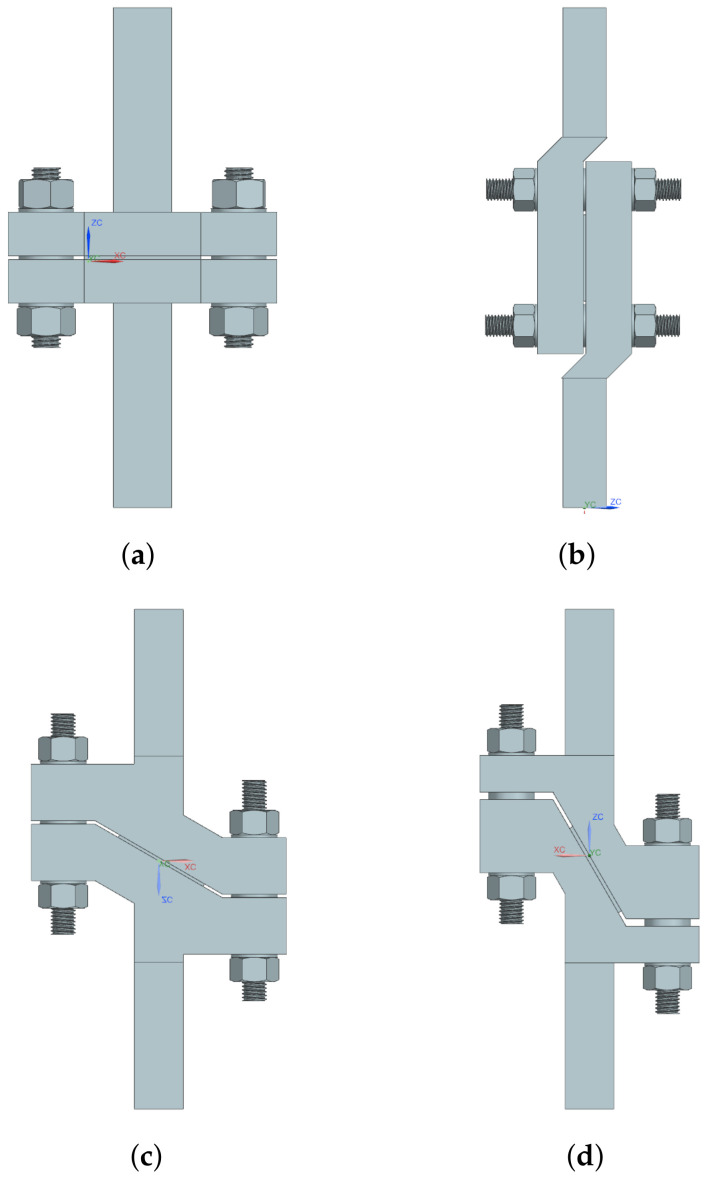
Four test tools are used for subjecting the sandwich specimens to LCs 1 and 3–5, presented in Figure 2. The test tool presented in (**a**) is used for transverse tensile loading, normal to the sandwich. In (**b**), the test tool used for loading the specimen in out–of–plane shearing is presented. In (**c**,**d**), the test tools for loading the sandwich 30° and 60° relative to the normal direction are presented.

**Figure 4 materials-14-04396-f004:**
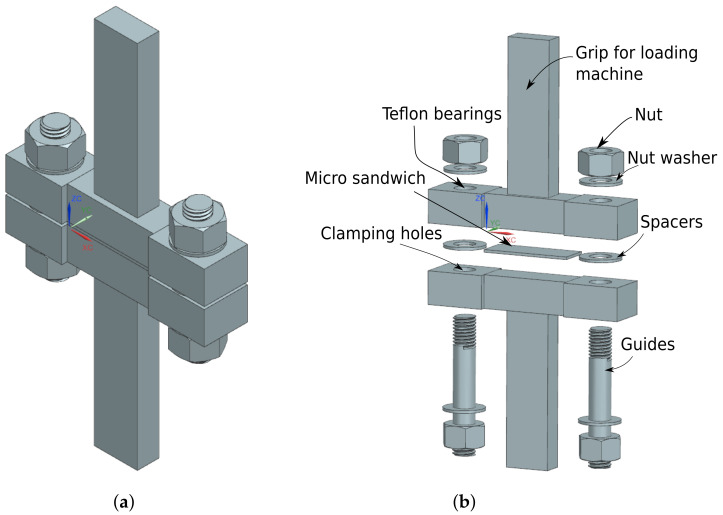
A perspective view of the test tool used for LC 1 is presented in (**a**). In (**b**), an exploded view of the test tool, with all constituents indicated, is presented.

**Figure 5 materials-14-04396-f005:**
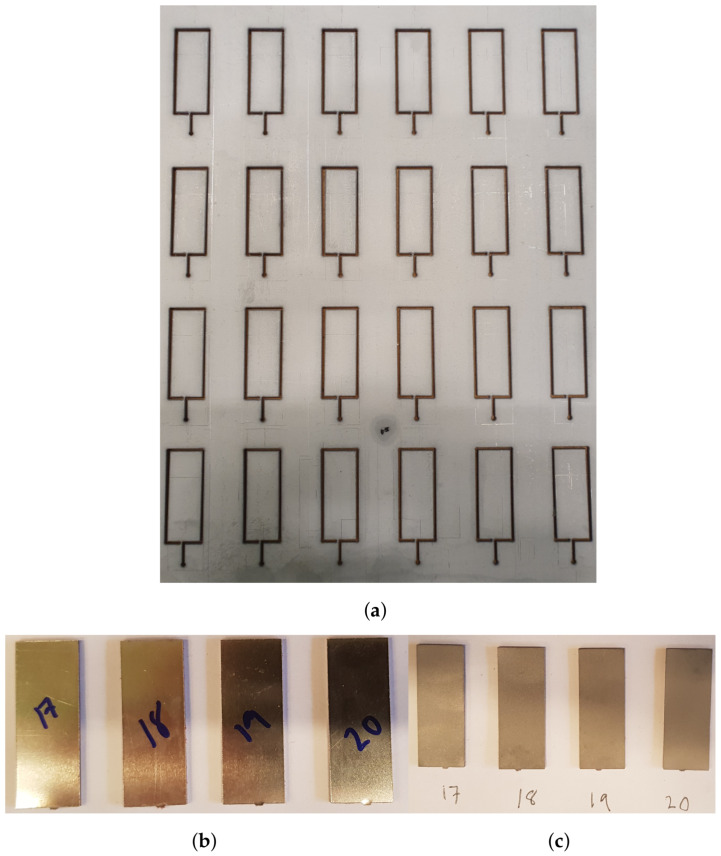
From the micro-sandwich sheet in (**a**), the specimens presented in (**b**) were obtained by water jet cutting. Before joining to the test tools, the test specimens were prepared using acetone and sand blasting, presented in (**c**).

**Figure 6 materials-14-04396-f006:**
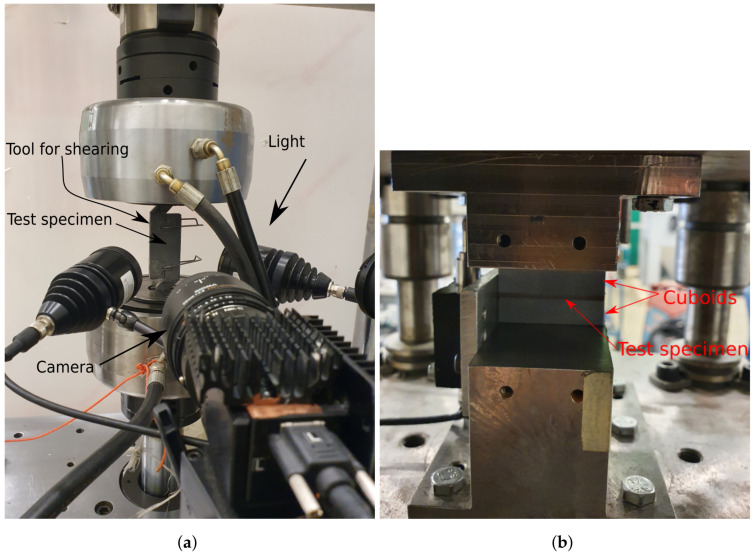
The experimental setup used for LCs 1 and 3–5 is presented in (**a**), with the test tool for shearing mounted in the machine. In (**b**), the experimental setup used of LC 2, i.e., compression, is presented. In both setups DIC was used.

**Figure 7 materials-14-04396-f007:**
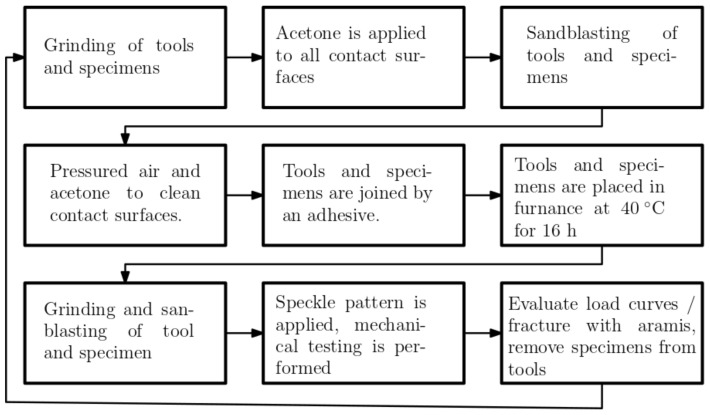
A flow chart for specimen preparation and testing.

**Figure 8 materials-14-04396-f008:**
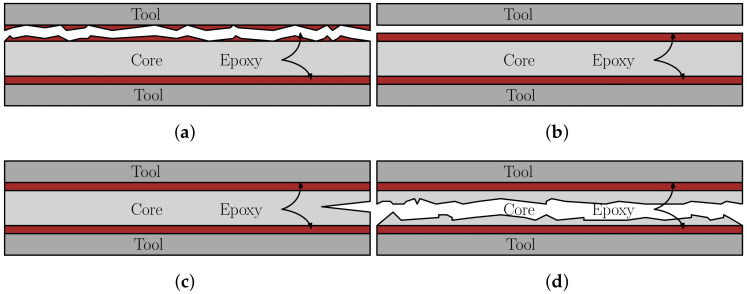
Four distinct failure modes were observed for LC 1. The failure modes presented in (**a**,**b**) corresponds to the performance of the bond, where a cohesive and adhesive failure are presented. In (**c**,**d**), the failure modes are related to the fracture of the core. In (**c**), the crack initiates from one of the edges and grows as loading proceeds. In (**d**), a brittle fracture occurs simultaneously over the specimen width.

**Figure 9 materials-14-04396-f009:**
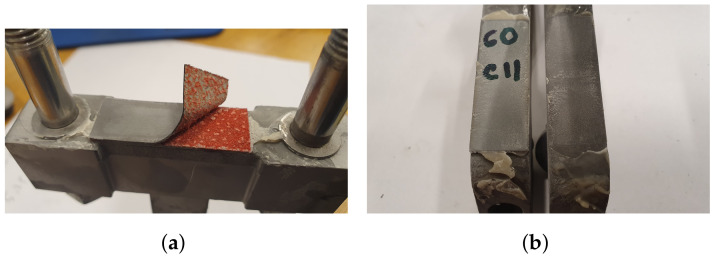
A mixed mode failure is presented in (**a**), where both a core failure and an adhesive failure has occurred. In (**b**), an adhesive failure is presented.

**Figure 10 materials-14-04396-f010:**
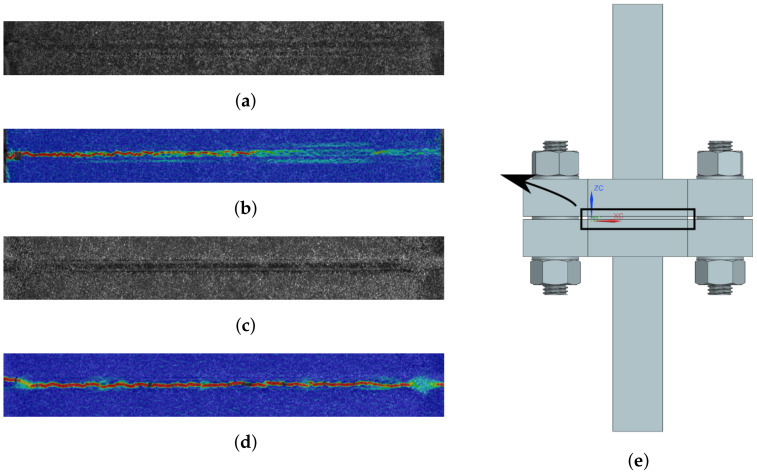
Two types of core fracture are compared in the figure: tearing and brittle fracture. (**a**,**c**) contain the speckle pattern, whereas (**e**) indicates, approximately, analyzed area. In (**b**), the strain field of the core is presented with strain localized near the left edge due to tearing. In (**d**), the strain is evenly distributed over the core, causing a brittle failure.

**Figure 11 materials-14-04396-f011:**
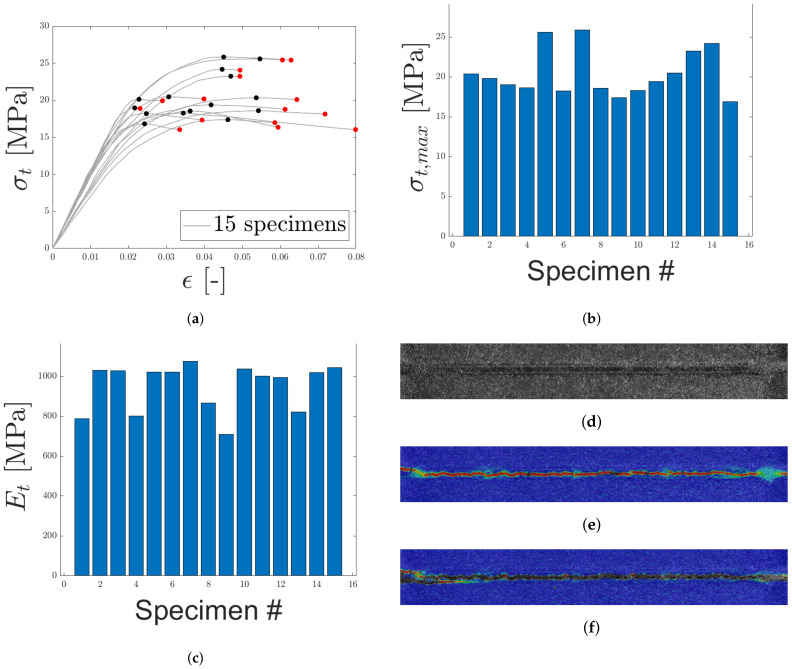
The uniaxial, engineering stress-strain response for LC 1 is presented in (**a**) with maximum stress and fracture marked by black and red, respectively. In (**b**,**c**), maximum uniaxial stress and equivalent Young’s modulus are presented, respectively. (**d**) contains the speckle pattern for one of the specimens. In (**e**,**f**), the strain field along the loading direction (before and after fracture), obtained using DIC, is presented.

**Figure 12 materials-14-04396-f012:**
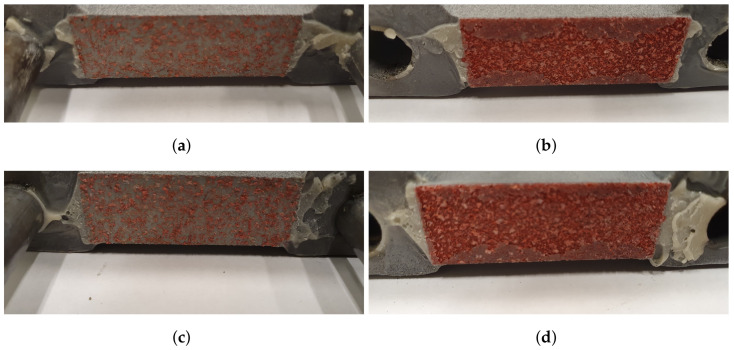
The fractured cores of two specimens, subjected to LC 1, are presented. (**a**,**c**) contain the lower skins and (**b**,**d**) contain the upper skins.

**Figure 13 materials-14-04396-f013:**
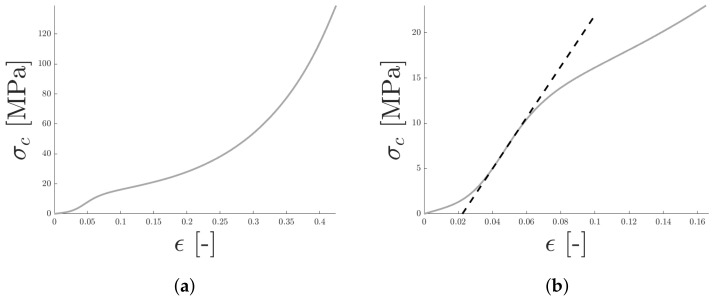
A typical loading response for uniaxial compression is seen in panel (**a**). The initial, nonlinear contact response was removed by extrapolating, using the equivalent Young’s modulus as illustrated in panel (**b**).

**Figure 14 materials-14-04396-f014:**
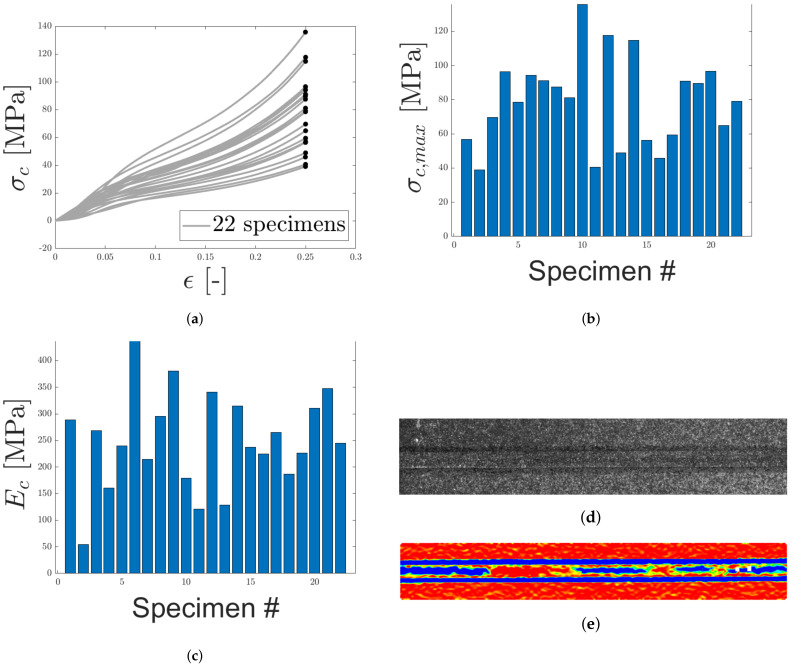
The egineering stress-strain response for LC 2 (uniaxial compression) is presented in (**a**). In (**b**,**c**) maximum stress and equivalent Young’s modulus are presented, respectively. (**d**) contains the speckle pattern for one of the specimens. In (**e**), the strain field along the loading direction, obtained using DIC, is presented.

**Figure 15 materials-14-04396-f015:**
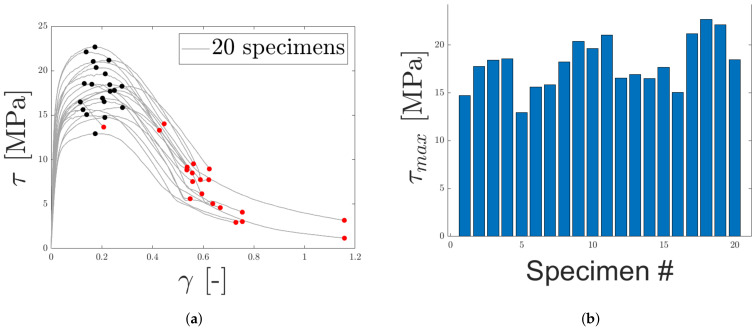
The engineering stress–strain response for LC 3 (out of plane shear) is presented in (**a**), with maximum stress and strain marked with black and red, respectively. In (**b**,**c**) maximum stress and equivalent shear modulus are presented. (**d**) contains the speckle pattern for one of the specimens. In (**e**,**f**), the strain field along the loading direction (before and after fracture), obtained using DIC, is presented.

**Figure 16 materials-14-04396-f016:**
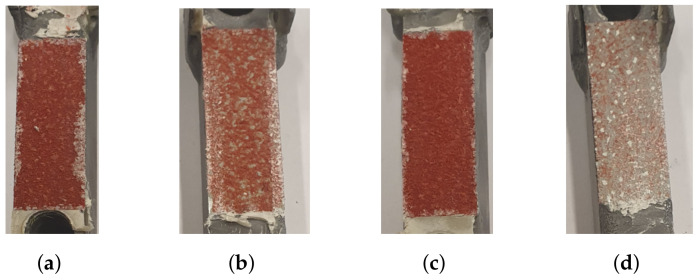
The fractured cores of two specimens, subjected to LC 3, are presented. (**a**,**c**) contain the lower skins and (**b**,**d**) contain the upper skins.

**Figure 17 materials-14-04396-f017:**
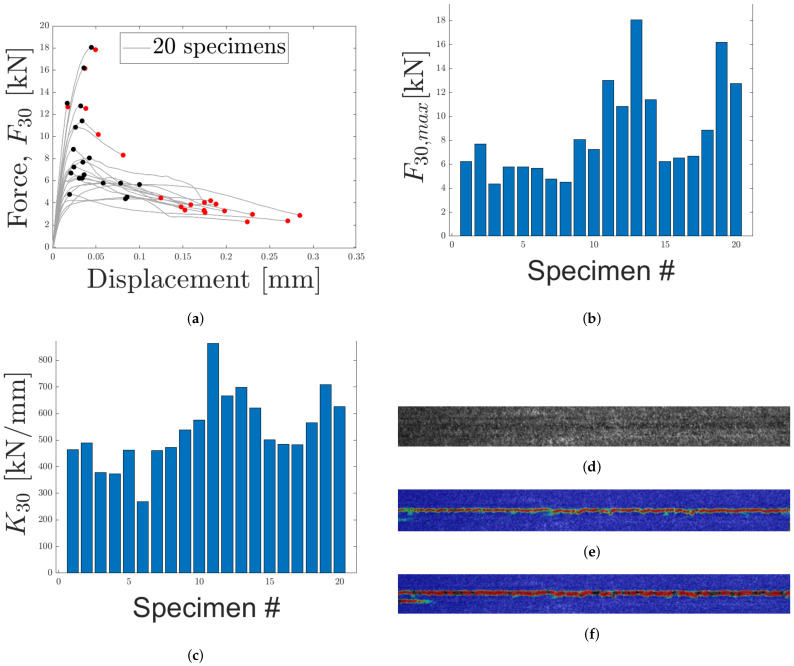
The force–displacement response for LC 4 (30° off axis loading) is presented in (**a**), with maximum force and displacement marked with black and red, respectively. In (**b**,**c**), maximum force and stiffness are presented. (**d**) contains the speckle pattern for one of the specimens. In (**e**,**f**), the strain field along the loading direction (before and after fracture), obtained using DIC, is presented.

**Figure 18 materials-14-04396-f018:**
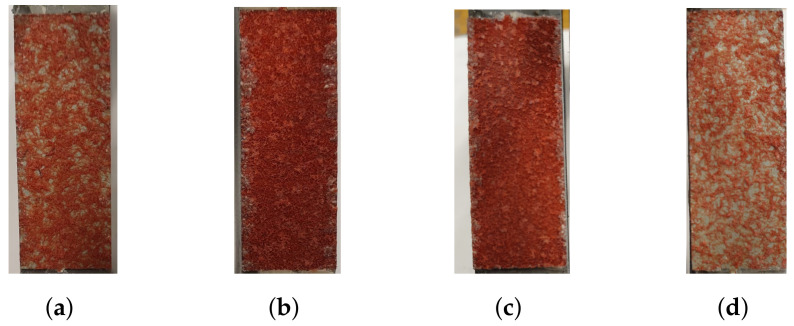
The fractured cores of two specimens, subjected to LC 4, are presented. (**a**,**c**) contain the lower skins and (**b**,**d**) contain the upper skins.

**Figure 19 materials-14-04396-f019:**
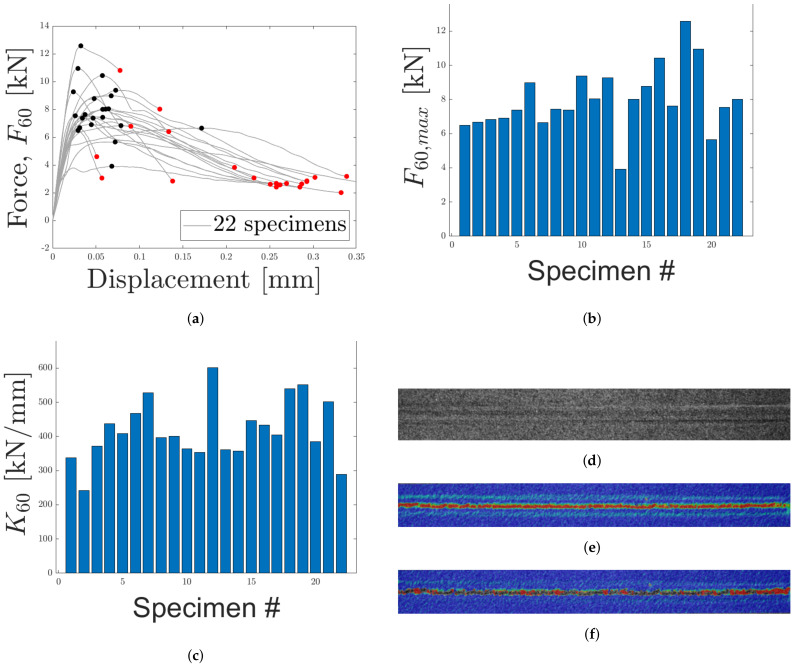
The force–displacement response for LC 5 (60° off axis loading) is presented in (**a**). In (**b**,**c**) maximum force and stiffness are presented. (**d**) contains the speckle pattern for one of the specimens. In (**e**,**f**), the strain fields along the loading direction (before and after fracture), obtained using DIC, are presented.

**Figure 20 materials-14-04396-f020:**
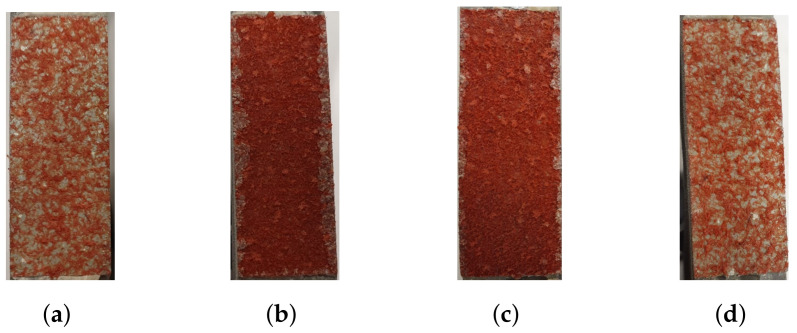
The fractured cores of two specimens, subjected to LC 5, are presented. (**a**,**c**) contain the lower skins and (**b**,**d**) contain the upper skins.

**Figure 21 materials-14-04396-f021:**
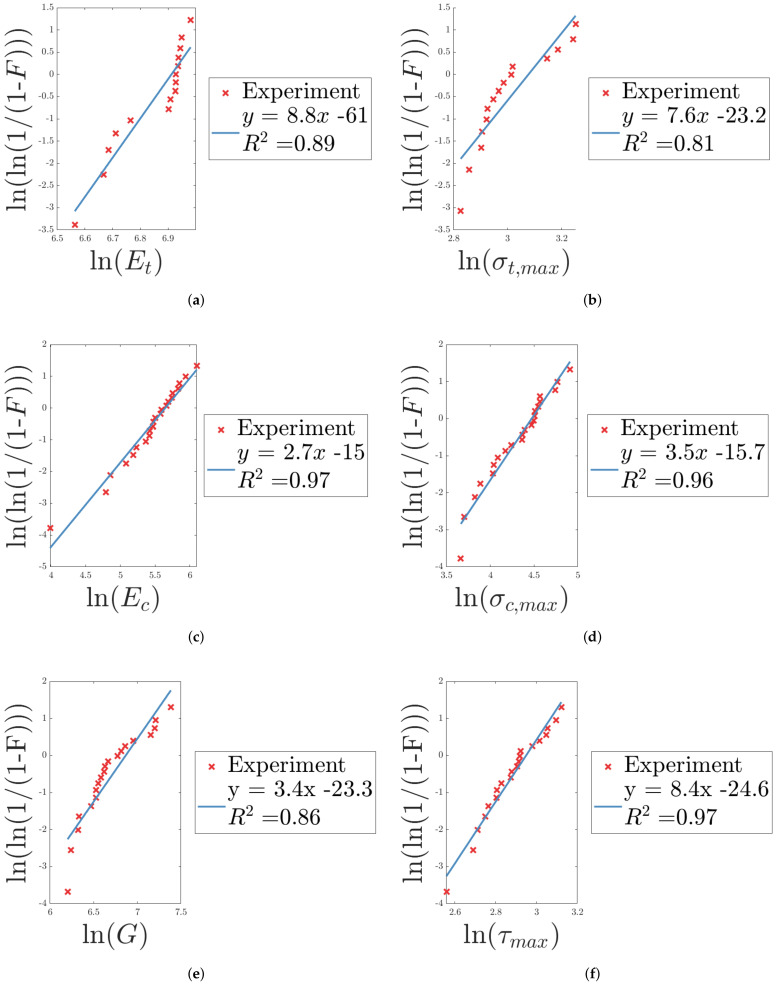
(**a**–**j**) contain the linearization (and goodness-of-fit) of the experimental data for each of the parameters in Table 4, from which the Weibull parameters are determined, according to the theory of Section 3.

**Figure 22 materials-14-04396-f022:**
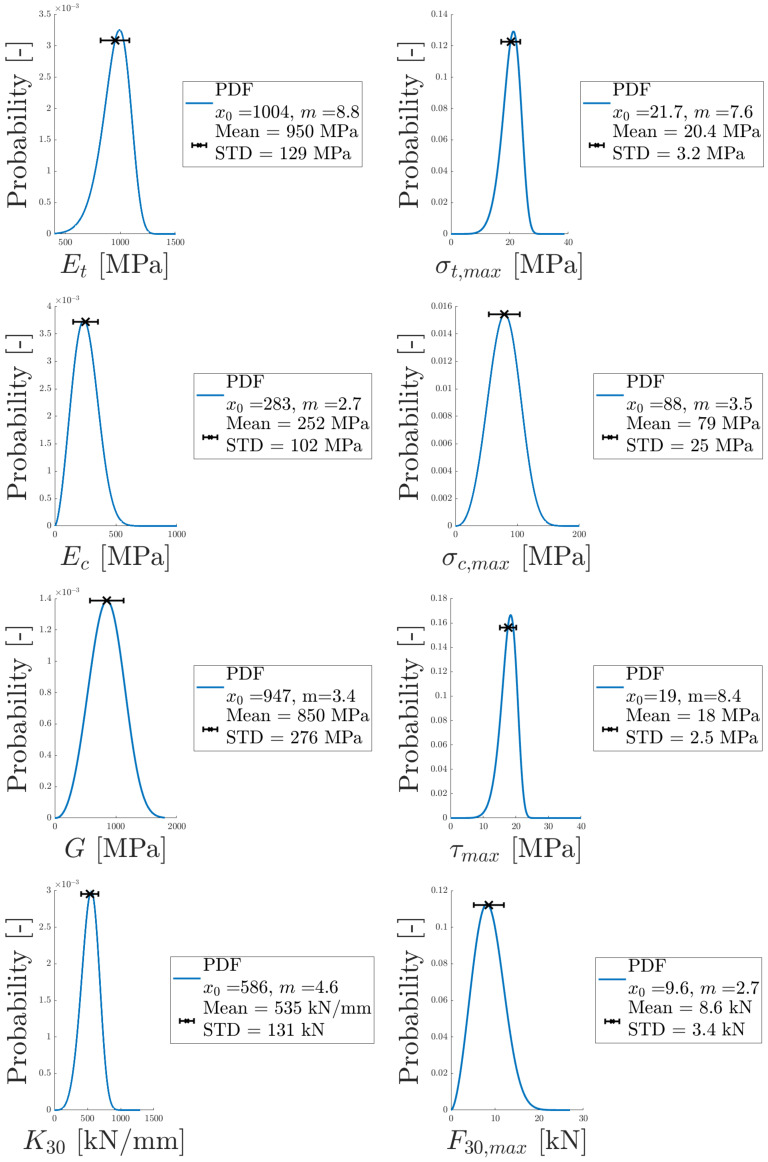
Probability distributions, obtained from the determined Weibull parameters, are presented for each parameter in Table 4.

**Figure 23 materials-14-04396-f023:**
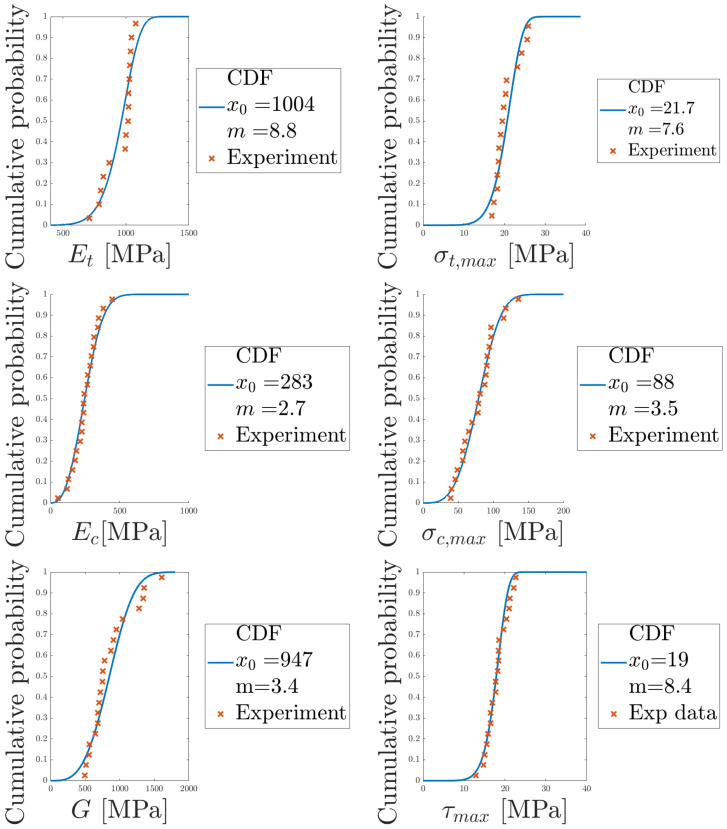
Cumulative density functions, obtained from the determined Weibull parameters, are presented for each parameter in Table 4.

**Figure 24 materials-14-04396-f024:**
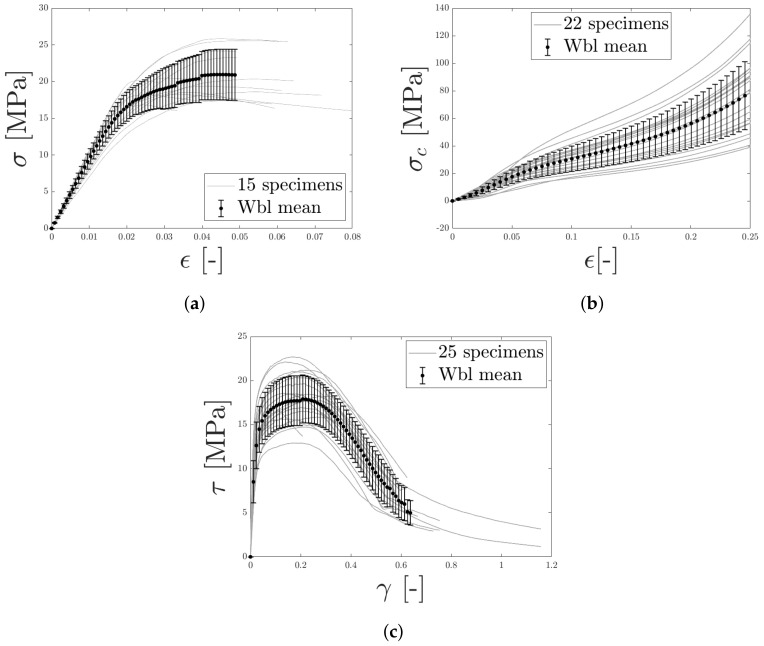
For LCs 1–3, a Weibull mean curve was computed and the standard deviation indicated. In (**a**,**b**), the transverse stress–strain response of tensions and compression are presented, respectively, whereas (**c**) contains the response from out–of–plane shear.

**Table 1 materials-14-04396-t001:** Material data for the CC030030PC-120 TMK-T7609 configuration of Hybrix^TM^ used in the study. *E* refers to Young’s modulus, ν to Poisson’s ratio, $\sigma_{y}$ to the initial yield stress, $\rho_{rel}$ to the relative density, and *t* corresponds to the thickness.

		*E* [GPa]	ν	$\sigma_{y}$ [GPa]	$\rho_{rel}$	*t* [mm]
Core	Polyamid + Adhesive					0.6
Skins	Carbon steel, DC04	200	0.3	0.2	1	0.3

**Table 2 materials-14-04396-t002:** The load cases used in the present study are summarized in the table.

Load Case	Loading Type
1	Transverse tension, normal to the micro-sandwich.
2	Transverse compression, normal to the micro-sandwich.
3	Out–of–plane shear.
4	Off–axis loading, 30°.
5	Off–axis loading, 60°.

**Table 3 materials-14-04396-t003:** Parameters and curves to be determined in the study.

Parameters/Curves	Description
$E_{t}$	Equivalent Young’s modulus in tension (normal to the sandwich).
$\sigma_{t,max}$	Maximum normal stress in tension.
$\sigma_{t}$ vs. $\epsilon_{t}$	Stress–strain curve (tension)
$E_{c}$	Equivalent Young’s modulus in compression (normal to the sandwich).
$\sigma_{c,max}$	Maximum normal stress in compression.
$\sigma_{c}$ vs. $\epsilon_{c}$	Stress–strain curve (compression)
*G*	Equivalent Shear modulus (out-of-plane)
$\tau_{max}$	Maximum shear stress.
τ vs. γ	Stress–strain curve (shear)
$K_{30}$	Stiffness for 30° off-axis loading
$F_{30,max}$	Maximum force for 30° off-axis loading
$F_{30}$ vs. $\delta_{30}$	Force vs. displacement for 30° off-axis loading
$K_{60}$	Stiffness for 60° off-axis loading
$F_{60,max}$	Maximum force for 60° off-axis loading
$F_{60}$ vs. $\delta_{60}$	Force vs. displacement for 60° off-axis loading

**Table 4 materials-14-04396-t004:** Parameters included in the statistical analysis.

Parameters	Description
$E^{t}$	Equivalent Young’s modulus in tension
$\sigma_{max}^{t}$	Maximum stress in tensions
$E^{c}$	Equivalent Young’s modulus in compression
$\sigma_{max}^{c}$	Maximum stress in compression
*G*	Equivalent shear modulus
$\tau_{max}$	Maximum shear stress
$K_{30}$	Stiffness 30° off-axis loading
$F_{30}$	Maximum force 30° off-axis loading
$K_{60}$	Stiffness 60° off-axis loading
$F_{60}$	Maximum force 60° off-axis loading

**Table 5 materials-14-04396-t005:** Summaryof results for LCs 1–5. ${\bar{x}}_{Wbl}$ corresponds to the Weibull mean value of the parameters and *R* is the correlation coefficient.

Load Case	Parameters	${\bar{x}}_{Wbl}$	*R*
1	$E_{t}$ [MPa]	950	0.26
	$\sigma_{t,max}$ [MPa]	20.4	
2	$E_{c}$ [MPa]	252	0.37
	$\sigma_{c,max}$ [MPa]	79	
3	*G* [MPa]	850	0.83
	$\tau_{max}$ [MPa]	18	
4	$K_{30}$ [kN/mm]	535	0.82
	$F_{30,max}$ [kN]	8.6	
5	$K_{60}$ [kN/mm]	417	0.51
	$F_{60,max}$ [kN]	8	

## Data Availability

The data that support the findings of this study are available from the corresponding author, S.H., upon reasonable request.

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
