# Peer review of "Novel Methodology for Experimental Characterization of Micro-Sandwich Materials"

_materials, 2021, doi:10.3390/ma14164396_

Round 1

Reviewer 1 Report

Comments to the Author:

In this manuscript, the authors present experimental methods to test the out of plane mechanical properties of HybrixTM by joining the material to test tools developed for this purpose. Compared to known work, this is a relatively more straightforward and direct methodology for micro-sandwich materials testing. However, the significance of this work should be better emphasized in order to attract a wider readership. Detailed results and discuss were included, however, the authors should consider the following issues:

  1. The figure captions and descriptions need some editing of its English language to present the figure more clearly, especially those with multiple pictures.

  1. Details of the instruments and methods (microscope, camera, magnification etc) need to be listed.

  1. Even though the manuscript claim the novelty of the methodology, this work is more like a continuation of work by Pimentel et al. The authors can elaborate more on how this methodology can be applied on broader range of materials, other than HybrixTM.

  1. The description about sample prep and experiment procedures should be more precise. The test tools are not clear in Figure 6, if they were loaded after curing with the specimen. The authors also need to be clear when referring to the adhesive used to join the specimen to the test tool and the adhesive between skin and core. The amount/thickness of Araldite 2015-1 adhesive used was not specified. Does adding a second adhesive affect the test results of the sandwich material? Especially when the test tool and joint adhesive can act like a second layer of the ‘sandwich’ bearing force, in which case the specs of the test tool should matter.

  1. Even though the displacements of the adhesives and test tool can be negligible, the skin needs to be considered. In which case, the mechanical properties tested are of whole material tested, not just the sandwich core, especially from the force-displacement curves. This may also reduce the significance of this work.

  1. What is the purpose of showing speckle patterns in figure 10-17, besides DIC images? No discussion about these images is included Was the speckle pattern applied on the side faces after joined to the tool? Figure 14f is missing.

Reviewer 2 Report

Novel methodology for experimental characterization of micro-sandwich panels – S. Hammarberg et al.

General comments: This paper presents experimental techniques to characterize mechanical properties of micro-sandwich materials. In particular, out-of-plane material constants were evaluated. The paper is well written and results are presented in a clear and concise fashion. The paper may be accepted in its current form. My specific comments are provided below.

Specific comments:

  • Use Italicize “t” for thickness.
  • Perhaps, Figure 1 and sub-figures can be described separately. For instance (a) Conceptual description…., (b) microscopic photo of …. , and (c) fibrous core …...
  • Line 173: The authors state that the core was subjected to mixed-mode stress state. How did the authors determine the mode-mixity?
  • Please provide a table listing various load-cases, LC 1-5 along with their loading type for ease of reference.
  • Is there any rationale behind selection of two-parameter individual Weibull over Joint Weibull method?
  • Section 8.1 title is misleading and may be re-titled as “Failure Surface Analysis or Characterization”.
  • Figure 8: (a) and (b) refers to adhesive characterization regardless if the failure is cohesive or adhesive. The study must indicate the preferred failure mode. For sandwich coupons in other industry the interface failure location & further progression is paramount (e.g. refer to Berggreen and Saseendran).
  • Are there any scaling studies being planned to characterize various LCs specimen designs? For micro-sandwich cases presented here, it this reviewer’s opinion that a sizing study must be conducted to reduce data scatter. In addition, such a study will better help in understanding the observed failure modes.
